

# Decoupling the response of an estuarine shrimp to architectural components of habitat structure

Jeffrey A. Crooks[1,2], Andrew L. Chang[2] and Gregory M. Ruiz[3]

[1] Tijuana River National Estuarine Research Reserve, Imperial Beach, CA, United States
[2] Smithsonian Environmental Research Center, Tiburon, California, United States
[3] Smithsonian Environmental Research Center, Edgewater, Maryland, United States

## ABSTRACT

In order to explore biotic attraction to structure, we examined how the amount and arrangement of artificial biotic stalks affected responses of a shrimp, *Palaemon macrodactylus*, absent other proximate factors such as predation or interspecific competition. In aquaria, we tested the effect of differing densities of both un-branched and branched stalks, where the amount of material in the branched stalk equaled four-times that of the un-branched. The results clearly showed that it was the amount of material, not how it was arranged, that elicited responses from shrimp. Also, although stalks were not purposefully designed to mimic structural elements found in nature, they did resemble biogenic structure such as hydroids, algae, or plants. In order to test shrimp attraction to a different, perhaps more unfamiliar habitat type, we examined responses to plastic "army men." These structural elements elicited similar attraction of shrimp, and, in general, shrimp response correlated well with the fractal dimension of both stalks and army men. Overall, these results indicate that attraction to physical structure, regardless of its nature, may be an important driver of high abundances often associated with complex habitats.

## INTRODUCTION

The physical nature of habitats profoundly shapes resident biotic assemblages. Habitat-species relationships underpin basic ecological dynamics (*McCoy & Bell, 1991*; *Matias et al., 2010*; *Tokeshi & Arakaki, 2012*), and also have implications for ecosystem management, conservation, and restoration (e.g., *Crooks, 2002*; *Jiménez & Conover, 2001*; *Thrush & Dayton, 2002*; *Byers et al., 2006*; *St. Pierre & Kovalenko, 2014*; *Loke et al., 2014*). Despite (or perhaps because of) the ubiquity and importance of habitat-species interactions, there are considerable conceptual and terminological issues associated with even the most fundamental aspects of habitat structure (*Matias, Underwood & Coleman, 2007*; *Tokeshi & Arakaki, 2012*; *Kovalenko, Thomaz & Warfe, 2012*; *Loke et al., 2014*). In general terms, however, habitat structure is typically considered to be related to two elements: the presence of distinct structural types–habitat heterogeneity; and the absolute abundance and configuration of structural material–structural complexity

Corresponding author
Jeffrey A. Crooks, jcrooks@trnerr.org

(*Heck & Wetstone, 1977*; *McCoy & Bell, 1991*; *Sebens, 1991*; *Beck, 2000*). Although this basic framework provides a foundation for considering how the physical nature of habitats shapes communities, these two elements are often confounded in studies examining biotic responses to habitats, making it difficult to quantify the effects of either (*Beck, 2000*; *Matias, Underwood & Coleman, 2007*).

The lack of clarity related to habitat structure notwithstanding, numerous studies have demonstrated that "complex" (*sensu lato*) habitats tend to have more individuals and/or species than less complex habitats, a pattern that has been observed across various systems and at various spatial scales (e.g., *Krecker, 1939*; *MacArthur & MacArthur, 1961*; *Recher, 1969*; *Orth, 1973*; *Dean & Connell, 1987*; *Crooks, 2002*; *Thomaz et al., 2008*). Complex habitats can serve as predation refuges (*Crowder & Cooper, 1982*; *Everett & Ruiz, 1993*; *Warfe & Barmuta, 2004*), and ameliorate competition (*Sale, 1975*). The structure of habitats can interact with physical conditions such as wind or currents, modulating the supply of resources such as food (*Gutiérrez et al., 2011*) or altering the distribution of materials to which species might respond (e.g., olfactory cues, *Ferner, Smee & Weissburg, 2009*). Structural complexity may increase living space, especially for relatively small organisms, and the fractal dimension of habitats has been found to correlate positively with density and negatively with body size (*Gunnarsson, 1992*; *Morse et al., 1985*; *McAbendroth et al., 2005*). Increased densities and diversities within complex habitats also can be caused by net inward fluxes of individuals. In aquatic systems, planktonic larvae can act as largely passive particles, and settlement rates in habitats such as seagrass beds may be increased due to decreased water velocities (*Fonseca et al., 1982*).

Active behavioral choice is also a proximate mechanism for motile species, and given that behavior can mediate individual-ecosystem interactions, behavioral studies are emerging as a key way to explore responses to human-induced environmental change, including impacts of "novel" habitats (*Wright et al., 2010*; *Sih, Ferrari & Harris, 2011*). In aquatic systems, attraction to objects–any object, regardless of whether it is natural or artificial–is a well-known response of a variety of fish and motile invertebrates (*Mortensen, 1917*; *Carlisle, Turner & Ebert, 1964*; *Salazar, 1973*; *Druce & Kingsford, 1995*; *Ingólfsson, 1998*; *Castro, Santiago & Santana-Ortega, 2001*). Further, architectural complexity can play a role in determining magnitude of responses (*Dean & Connell, 1987*; *Hacker & Steneck, 1990*). A variety of lab and field experiments have examined behavioral responses to the structural complexity of habitats (e.g., *Bell & Westoby, 1986*; *Jeffries, 1993*; *Gee & Warwick, 1994*; *Robertson & Weis, 2007*). Habitat selection is often evaluated in the presence of predation pressure, either by design or as a consequence of field settings (*Crowder & Cooper, 1982*; *Everett & Ruiz, 1993*). Fewer studies have looked at habitat choice in the absence of factors such as predation pressure or food supply, but those that have indicate that the physical arrangement of materials alone can play an important role in shaping the distribution and abundance of motile organisms (*Stoner, 1980*; *Bell & Westoby, 1986*; *Hacker & Steneck, 1990*).

In this study, we examined attraction to the amount and arrangement of structural elements by a motile organism, the shrimp *Palaemon macrodactylus*. This grass shrimp is native to Asia, but has been introduced worldwide (*Micu & Niță, 2009*). It was first

reported in the San Francisco Bay in the 1950's (*Newman, 1963*), and is now one of the most common nektonic organisms in the brackish waters there (*Hatfield, 1985*; J. Crooks, 2000, personal observation). Like many motile macrofauna, this shrimp is often found associated with structure, with *Newman (1963)* noting that it was "frequently found in old tires, tin cans and other artificial structures as well as on pilings, walls and among rocks or calcareous tubes of *Mercierella*" (*Mercierella = Ficopomatus enigmaticus*, a non-native, "reef-creating" polychaete worm).

The goal of our work was to examine how the complexity and density of structural elements would drive behavioral responses of *P. macrodactylus*, while controlling for other proximate factors which might drive shrimp response (e.g., immediate predation risk and presence of food). This consisted of two experiments, one where shrimp could choose between arrays of structural elements in different configurations, and the other where we examined the response to a single array to assess the degree to which habitat complexity, as assessed with total surface area and fractal dimension, could account for position of shrimp in experimental tanks. The structural elements used in the experiments were plastic mesh cut into different configurations. While these artificial structures were not deliberately constructed to resemble any particular species, they did resemble various organisms in benthic communities (Fig. 1A), such as macroalgae, vascular plants, or hydroids. Other researchers have used similar "mimics" to examine responses of aquatic fauna (e.g., *Schneider & Mann, 1991*; *Jeffries, 1993*; *Bourget & Harvey, 1998*). In order to determine how shrimp would respond to perhaps more unfamiliar structural forms, we also created two "non-mimic" treatments, consisting of plastic "army men" (toy soldiers and equipment; Fig. 1B).

## METHODS

### Location

Asian grass shrimp *Palaemon macrodactylus* were collected from a floating dock at Black Point, where the Petaluma River enters north-western San Francisco Bay. The shrimp were abundant and easily collected with dip nets, and were transported to the greenhouse facility at Romberg Tiburon Center (San Francisco State University), where they were kept in holding tanks and fed daily with flake fish food. Average-sized shrimp, approx. 4 cm in length, were selected for use in the experiments, and individual shrimp were used only once (*Underwood, Chapman & Crowe, 2004*).

### Experimental setup

All experiments were conducted in 10-gallon aquaria (bottom dimensions = 50.8 cm × 25.4 cm) with a thin (1–2 cm) layer of sand covering the bottom. The 12 cm-long stalks were cut in two configurations from 1-mm diameter black plastic mesh with approximately 15 × 15-mm openings. Two levels of architectural complexity were used (Figs. 1A and 2). The un-branched stalk was a single strip of mesh with all side branches removed. The branched stalk was cut out of the mesh in such a way that crosspieces were left intact and one branched stalk had the same amount of material and surface area as four un-branched stalks. Two, four, eight, sixteen, and thirty-two stalk configurations

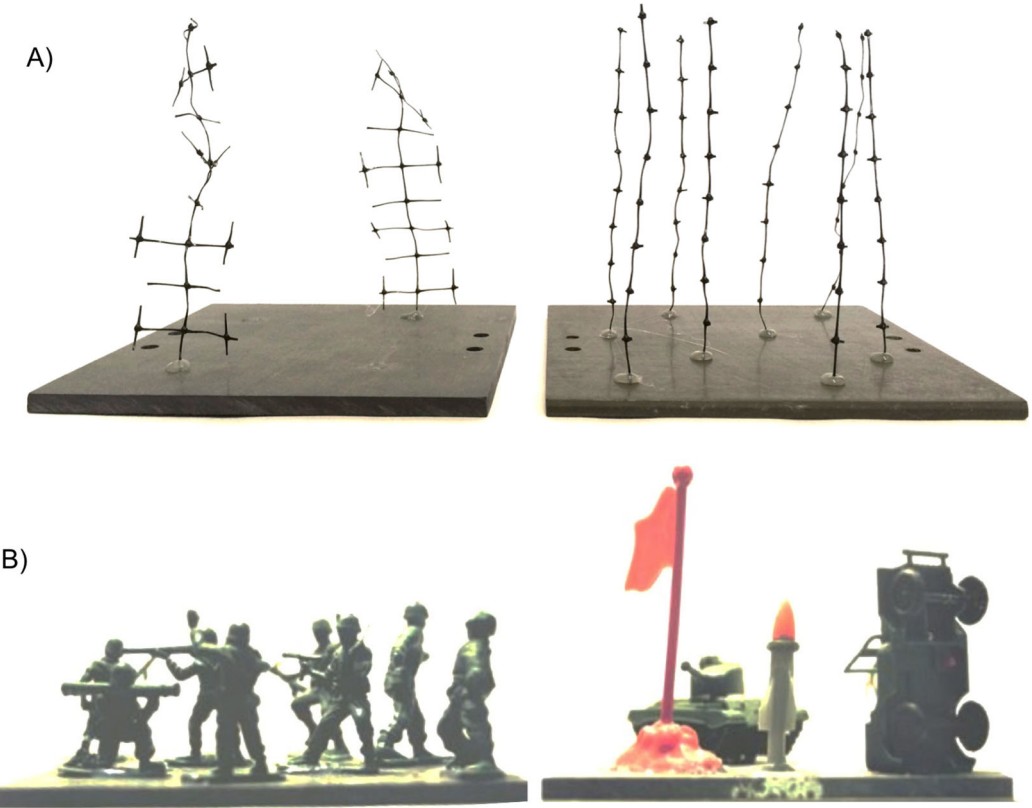

**Figure 1 Examples of experimental treatments.** (A) Unbranched and branched stalks, showing a configuration with equal surface area. (B) The novel, "non-mimic" habitat–plastic army men and equipment.

were used to assess shrimp responses to structure. Structural material was arrayed uniformly on the 14 × 14-cm PVC plates and fixed in place with hot melt glue. The two non-stalk treatments were created using arrays of toy plastic army men and army equipment (Fig. 1B). These treatments offered different structural types, with less resemblance to naturally-occurring biogenic structure. There were eight army men per plate, and four pieces of army equipment.

Panels were placed on the bottom of the tank with the surfaces buried so that only the stalks or toys were visible above the sand. The test area was the surface immediately above the entire 14 × 14-cm panel (Fig. 2). All tanks were kept within in a blackout tent to control light exposure, with illumination provided by overhead fluorescent lamps set on a timer with 12 h light/dark cycles. Individual tanks were wrapped on all vertical sides with opaque black plastic to prevent reflections and reactions to shrimp in neighboring tanks.

At the beginning of each trial run, five shrimp were placed in each tank and left to acclimate overnight. The next morning, observations were taken every half hour for 6 h, for a total of 13 observations. Observations were taken by lifting the opaque plastic cover from one long vertical side of each tank and counting the number of shrimp in the test area. Testing before the experiment identified this as the best method for observing shrimp without startling them, and during the course of the experiment we did not notice shrimp respond to our observations. This set-up was used in two experiments.
## Experimental Set-Up

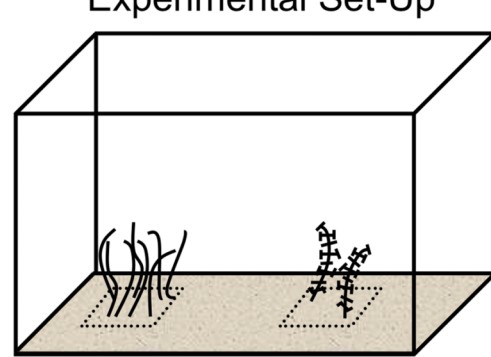

### Stalk Arrays
(per 14 x 14 cm test area)

| Total Surface Area (cm²) | Number of Stalks | |
|---|---|---|
| | Branched | Unbranched |
| 0.75 | | 2 |
| 1.5 | | 4 |
| 3.0 | 2 | 8 |
| 6.0 | 4 | 16 |
| 12.1 | 8 | 32 |
| 24.1 | 16 | |
| 48.2 | 32 | |

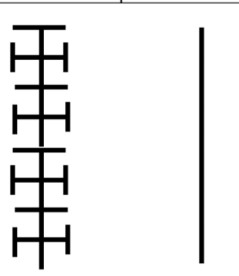

Choice Between Arrays

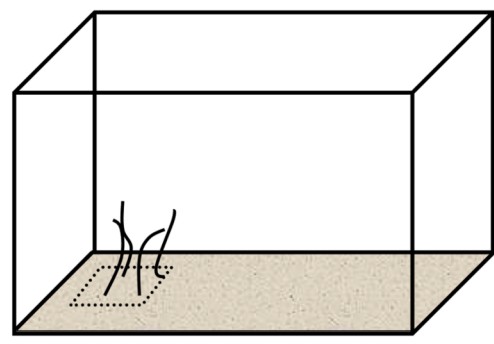

Response to Single Array

**Figure 2 The experimental design.** Stalks were constructed so that four un-branched stalks had the same surface area as one branched stalk. Two experiments were run, one where shrimp could choose between structural arrays on opposite sides of the tank, and one where shrimp responded to a single structural array on a randomly-selected side.

## Choice between two arrays

This experiment assessed shrimp responses when presented with structural arrays of varying architectural complexity on opposite sides of the tanks. Two 14 × 14-cm structural arrays (the test areas) were present in each tank, one on each side, which were randomly chosen (Fig. 2). The shrimp were offered four different pairings. There were two choices between arrays of equal stalk density: 8 un-branched vs. 8 branched, or 32 un-branched vs. 32 branched. There were also two choices with equal surface area of material: 2 branched vs. 8 un-branched, or 8 branched vs. 32 un-branched. The response variable was the average number of shrimp in the each 14 × 14-cm test area over the course of a trial run. The experiment was a Randomized Complete Block Design, blocked in time and with tanks with each treatment pairing running simultaneously. This was repeated 10 times, with a re-randomization of treatment allocation to tank each time.

## Response to a single array

This experiment examined shrimp responses to the presence of a single structural array of varying architectural complexity on one randomly-chosen side of each tank, with nothing on the other side. Treatments consisted of 5 densities of un-branched stalks and 5 densities of branched stalks. The response variable was the average number of shrimp in

the single $14 \times 14$-cm test area over the course of a trial run. The experiment was again blocked in time, with eight blocks and a re-randomization of treatment allocation to tank each time. Also, the two non-stalk treatments (plastic army men and army equipment) were included in each block.

### Analyses

All shrimp density data were log-transformed prior to analysis, and a constant of 0.077 was added to all values. This represents the smallest non-zero number that could be obtained (e.g., *Warton & Hui, 2011*): one shrimp in one test area during one observation period. In the *Choice Between Two Arrays* experiment, each pairing was analyzed separately using paired t-tests. For the *Response to a Single Array* experiment, the effects of stalk density, type, and block were analyzed as a three-factor ANOVA without replication. Comparisons between arrays of equal surface area were conducted using paired t-tests.

For the assessment of the effect of toys, the number of shrimp in the test area (out of the five total) were compared to what would be expected if shrimp were randomly distributed on the tank floor using a t-test (i.e., an average of 0.69 shrimp would be expected in the $14 \times 14$-cm test area, which constitutes 13.8% of the total tank bottom). Number of shrimp in the test area was also plotted against fractal dimension of each array type, including the toys. This was done using the program Fractal Dimension Calculator (http://paulbourke. net/fractals/fracdim/), which uses the box-counting method as an estimate of fractal dimension (*Morse et al., 1985*; *Jeffries, 1993*). In this procedure, a grid is superimposed upon a digitized image and the number of cells occupied by the image is calculated iteratively for progressively smaller cell sizes. Images for this analysis were taken from the sides of three replicate arrays for each array type. In order to assess if shrimp responded to the toys similarly as the stalks, we determined the degree to which individual points affected the relationship between fractal dimension and number of shrimp using Cook's D (*Fox, 1991*). A cutoff for "influential" points of $4/(n-k-1)$ was used (*Fox, 1991*), where $n = 12$ (number of observations) and $k = 1$ (number of independent variables), resulting in a value of 0.40.

## RESULTS

### Choice between two arrays

In the case where shrimp in a single tank had a choice between two arrays with differing structural arrangements, the total amount of material, and not how it was arranged, was the primary driver of shrimp response (Fig. 3). In tanks with test areas with equal amounts of material but differing stalk density, no statistically significant differences in number of shrimp were found ($P \geq 0.39$). At equal stalk density, however, shrimp preferred test areas with branched stalks, choosing areas with more structure over those with less (Fig. 3). This effect was particularly apparent at the highest stalk densities, with over 8 times as many shrimp found with branched stalks than with un-branched ($P < 0.001$).

### Response to a single array

When presented with a single array per tank, the quantity of structure again had marked effects on shrimp location within tanks (Fig. 4). Both stalk density and stalk type
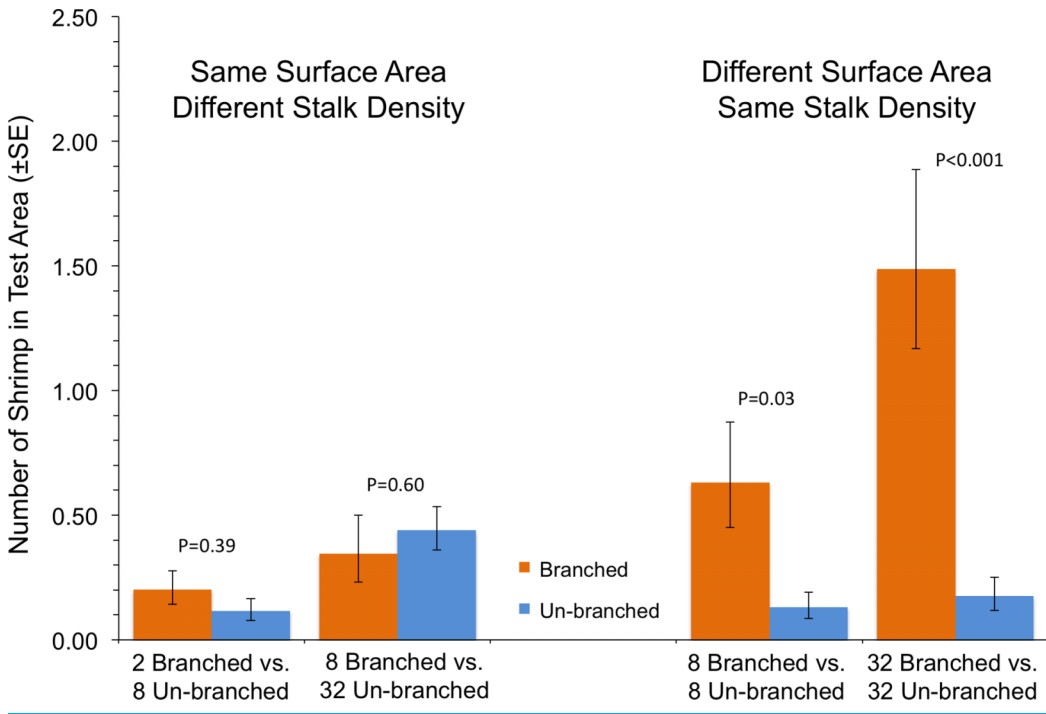

**Figure 3** Shrimp responses when given a choice between structural arrays with the same surface area or same stalk density.

(branched vs. un-branched) were statistically significant. For both stalk types, number of shrimp increased approximately 500% from the lowest to the highest stalk densities. However, there were no significant differences in shrimp densities in test areas with equal amounts of material but differing stalk density (Fig. 4–dashed lines), as also observed in the choice experiment (Fig. 3). In general, total stalk surface area, irrespective of stalk type, was an excellent predictor of number of shrimp within the test area (Fig. 5).

The shrimp also displayed marked responses to the non-stalk habitat treatments. The army men treatment had a mean density of 2.3 ($\pm$ 0.2) shrimp per 14 $\times$ 14-cm test area, and the army equipment treatment had a mean density of 1.6 ($\pm$ 0.4). Both of these are significantly different than what would be expected if shrimp were randomly distributed on the tank bottom (army men: $t_7 = 11.5$, P $<$ 0.001; army equipment: $t_7 = 3.22$, P $= 0.015$). Also, these shrimp densities are comparable to the two highest values found using branched stalk treatments.

Fractal dimension, as assessed with the box-counting method, was again an excellent predictor of number of shrimp within the test area (Fig. 6). Although stalks and toys represent quite different structural forms (as well as textures and colors), fractal dimension accounted for 93% of the variability to shrimp number ($R^2 = 0.93$, P $<$ 0.001). The values for Cook's D, which assesses outliers (or "influential points") in regressions, all fell below the cut-off of 0.4 (although the value for the army equipment approached this at D $= 0.37$). Also, for the stalk treatments, it is important to note that the amount of material was also very closely related to the fractal dimension of the arrays (logarithmic regression, $R^2 = 0.98$, P $<$ 0.001).

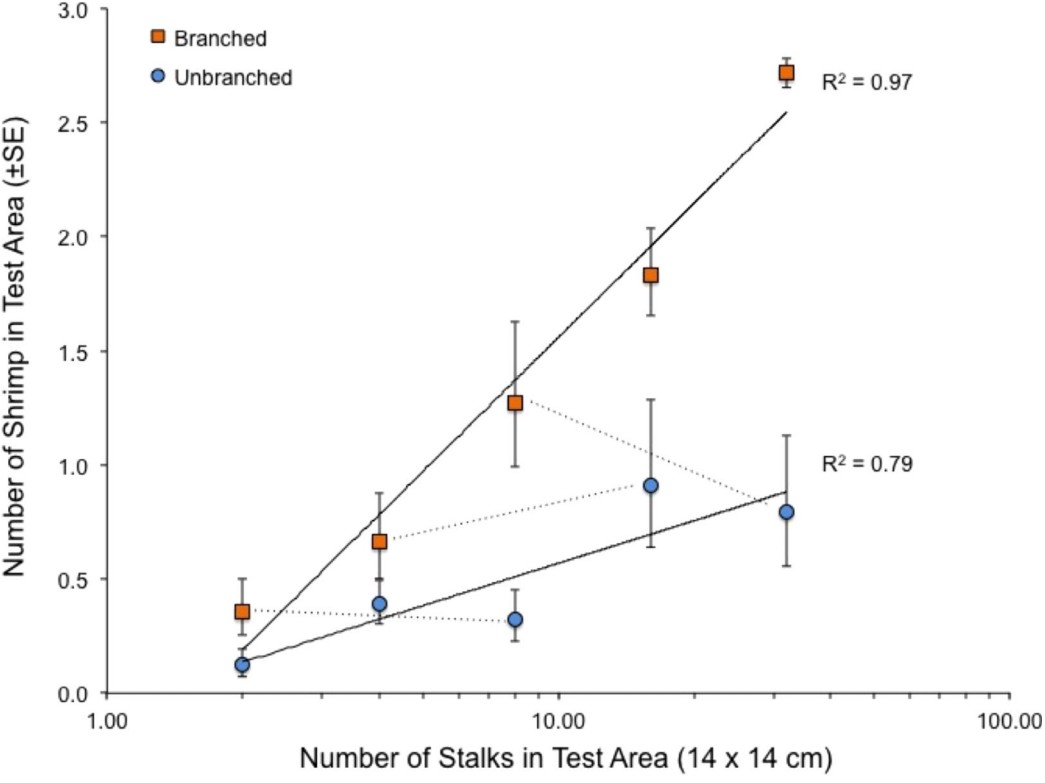

**Figure 4  Relationship between number of stalks and number of shrimp in the test area in the tanks with a single array.** Also shown with dashed lines are the comparisons between arrays with equal stalk surface area, which were all statistically non-significant (paired t-tests, 7 df, P-values > 0.34). See also Table 1.

## DISCUSSION

These experiments demonstrate that shrimp responded to the presence of structure in the absence of proximate factors such as predation pressure and food supply, and that an excellent predictor of shrimp location was the amount of material available. These findings support previous suggestions that surface area tends to be a good measure of complexity (*Stoner, 1980*; *Stoner & Lewis, 1985*; *Beck, 2000*). For example, *Attrill, Strong & Rowden (2000)* found that macro-invertebrate community structure in seagrass beds was associated with the amount of plant available, and *Dean & Connell (1987)* showed that intertidal invertebrates chose algal clumps of based on biomass rather than species identity. Not surprisingly, however, exceptions to this relationship are found, and morphologically different structures with constant surface areas have been shown to elicit differing responses (*Loke & Todd, 2016*), such as artificial macrophyte beds with many thin blades having higher fish abundance than beds with fewer, thicker blades (*Jenkins & Sutherland, 1997*). In other cases, surface area might underestimate the number of individuals and species found within increasingly complex habitats (*Hauser, Attrill & Cotton, 2006*), or, conversely, there might be negative relationships between amount of material and invertebrate communities (*Kelaher, 2003*). Together, these emphasize that other processes are at work beyond just surface area, and it is necessary to consider

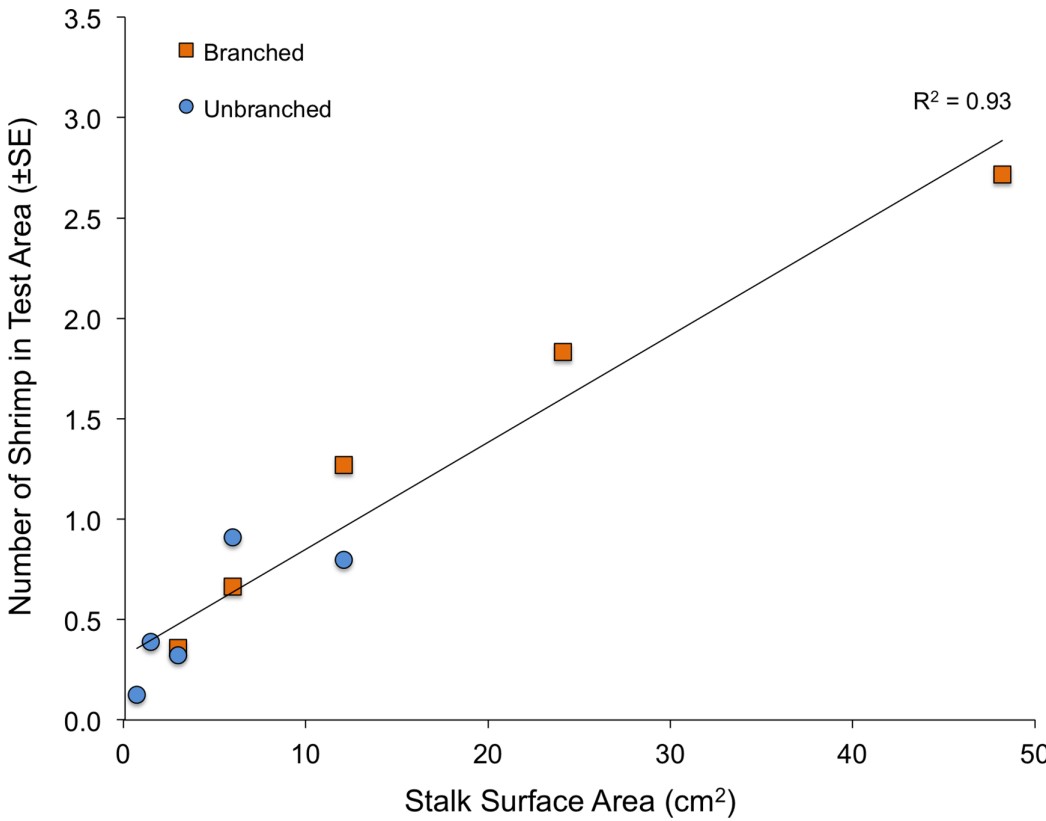

**Figure 5** Relationship between total stalk surface area and number of shrimp in the test area in tanks with a single array.

**Table 1** ANOVA table for the choice experiment, examining the average number of shrimp in the test areas with differing combinations of stalk density and type.

| Source of variation | df | MS | F | P-value |
| --- | --- | --- | --- | --- |
| Density (n = 5) | 4 | 1.37 | 12.2 | < 0.001 |
| Stalk type: unbranched vs. branched (n = 2) | 1 | 2.75 | 24.4 | < 0.001 |
| Block (n = 8) | 7 | 0.21 | 1.9 | 0.11 |
| Block * Density | 28 | 0.06 | 0.5 | 0.95 |
| Block * Stalk type | 7 | 0.26 | 0.2 | 0.98 |
| Density * Stalk type | 4 | 0.08 | 0.7 | 0.60 |
| Residual | 28 | 0.11 | | |

factors such as scale and environmental context (*Jenkins, Walker-Smith & Hamer, 2002*; *Kovalenko, Thomaz & Warfe, 2012*; *Matias, 2013*).

Fractal dimension also was a powerful predictor of shrimp response. Although it is possible that shrimp were also responding to differing cues (such as material or color), fractal dimension explained much of the variability in shrimp response to the two stalk types as well as army toys (Fig. 6). Like other experiments that explicitly attempted comparisons of fractal dimension and surface area (*Beck, 2000* and references therein), our results have similarly resulted in a limited ability to recommend one over the other given

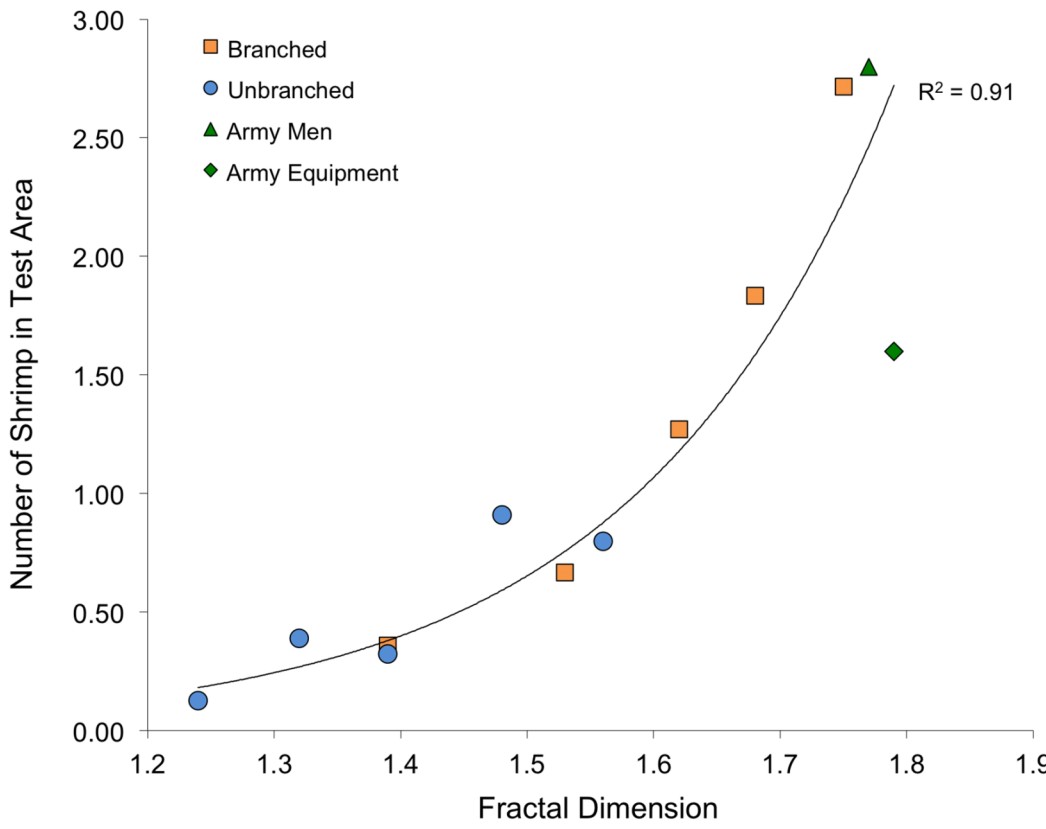

**Figure 6** Relationship between the fractal dimension of the stalk arrays and toy treatments with and number of shrimp the test area.

that surface area and fractal dimension were correlated. It is worth noting, though, that fractal dimension, assessed with the box-counting method, is easier to assess than surface area for irregular structures (*Beck, 2000*), but can be difficult to apply in the context of applied conservation or restoration activities (*Loke et al., 2014*). Although several studies have shown that fractal surfaces offer more usable space for smaller organisms than larger ones, producing a positive relationship between abundance and complexity (*Morse et al., 1985*; *Shorrocks et al., 1991*), this argument does not apply here as the shrimp were all comparably-sized (approximately 4-cm).

Our results also agree with other studies indicating that behavioral choice is an important factor shaping the distribution and abundance of motile organisms (e.g., *Stoner, 1980*; *Rooke, 1984*; *Bell & Westoby, 1986*). In the laboratory setting of these experiments, shrimp were attracted to branched and un-branched structures that generally resembled materials such as hydroids, algae, or plants. In addition, the shrimp responded to forms that might be less typical of natural systems, the plastic army toys. Together, these responses suggest that for these shrimp choice is largely based on the presence of structure of any sort, and not just a response to familiar forms.

Despite the potentially strong effect of choice on patterns of distribution and abundance, more work is needed on underlying mechanisms, including habitat choice in

response to specific behaviors being displayed, such as mating, navigation, feeding, or predator escape (*Shumway, 2008*; *Kovalenko, Thomaz & Warfe, 2012*; *Roever et al., 2014*). Many inter-related factors could potentially drive attraction to structure, including thigmotactic movement oriented by contact or touch (*Olyslager & Williams, 1993*; *Uryu, Iwasaki & Hinoue, 1996*; *Webster & Laland, 2011*), or structure offering a visual reference point for activities such as breeding or feeding (e.g., *Dagorn & Fréon, 1999*). For some organisms, such as birds and mammals, it has been suggested that psychological factors, such as aversion or attraction to novel conditions, serve to structure behaviors and distributional patterns (*Greenberg, 1984*). Also, although predators were not immediately involved in our experiments, there is likely recognition that complex habitats may afford escape from predation (*Bell & Westoby, 1986*; *Shumway, 2008*), and species might alter behavior according to the perceived protective value of habitats even in the absence of immediate predation risk (*Ingrum, Nordell & Dole, 2010*).

The response of species to the physical nature of habitats, and the general tendency to choose more complex over simpler structures, has a number of implications for applied ecology. For example, a better understanding of the behavioral responses to structure can improve design of artificial reefs (*Gratwicke & Speight, 2005*; *Perkol-Finkel, Shashar & Benayahu, 2006*) and fish aggregating devices (*Dagorn & Fréon, 1999*; *Castro, Santiago & Santana-Ortega, 2001*; *Girard, Benhamou & Dagorn, 2004*). The generality of behavioral responses to structure also allows prediction of the effects of habitat-modifying exotic species, as invasive ecosystem engineers may create unfamiliar, novel habitat types that tend to benefit at least some resident biota regardless of the exact type of structure being created (*Crooks, 2002*). Clearly, there will be exceptions to the pattern of positive relationships between complexity and species responses (*Crooks, 2002*; *Gutiérrez & Iribarne, 2004*). For example, complex habitats can impair visual fields (*Rilov et al., 2007*) or interfere with foraging, such as the invasive alga *Caulerpa* interfering with feeding of native mullet (*Levi & Francour, 2004*).

Given the potential importance of the recognition and response of organisms to habitat complexity, more research is needed on this subject (*Shumway, 2008*). In general, it will be valuable to examine habitat preferences in the absence of other extrinsic factors in a wide range of species, as understanding such interspecific differences may help explain community-level patterns (e.g., *Greenberg, 1984*). For *P. macrodactylus*, which has been shown to demonstrate a high degree of correlation between habitat complexity and magnitude of response (this study), it would be interesting to determine whether responses are innate rather than learned (which might be accomplished by examining lab-reared versus field-caught individuals), as well as to conduct experiments with varying densities of conspecifics in order to determine the potential role of density-dependent, intraspecific interactions in shaping distribution. It also would be valuable to compare results of artificial structures to natural ones, including carefully-controlled field experiments to explore shrimp responses in the presence of other factors that would affect distribution in natural settings (*Underwood, Chapman & Crowe, 2004*).

## CONCLUSIONS

The results of these and other experiments demonstrate that attraction to physical structure in the absence of proximate drivers can explain patterns of increased densities within complex habitats. However, it is clear that habitat structure affects resident biota in many other fundamental ways, including active and passive accumulation of individuals (*Fonseca et al., 1982*), amelioration of competition (*Sale, 1975*) and environmental conditions (*Bruno & Bertness, 2001*), and predator avoidance (*Everett & Ruiz, 1993*). As many of these factors will tend to promote higher densities within complex habitats, it can be difficult to tease out the relative importance of each when all are potentially operating. Nevertheless, it is likely this synergism that drives attraction to complex structure.

## ACKNOWLEDGEMENTS

We thank the staff at the Romberg Tiburon Center, San Francisco State University, for helping to facilitate this work.

### Funding

Portions of this work were funded by the Smithsonian Institution and the Maryland Sea Grant Program (R/IS-10). The funders had no role in study design, data collection and analysis, decision to publish, or preparation of the manuscript.

### Grant Disclosures

The following grant information was disclosed by the authors:
Smithsonian Institution and the Maryland Sea Grant Program: R/IS-10.

### Competing Interests

The authors declare that they have no competing interests.

### Author Contributions

- Jeffrey A. Crooks conceived and designed the experiments, performed the experiments, analyzed the data, wrote the paper, prepared figures and/or tables, reviewed drafts of the paper.
- Andrew L. Chang conceived and designed the experiments, performed the experiments, analyzed the data, wrote the paper, prepared figures and/or tables, reviewed drafts of the paper.
- Gregory M. Ruiz conceived and designed the experiments, contributed reagents/materials/analysis tools, wrote the paper, reviewed drafts of the paper.

### Data Deposition

The raw data has been supplied as Supplemental Dataset Files.

## Supplemental Information

Supplemental information for this article can be found online at http://dx.doi.org/10.7717/peerj.2244#supplemental-information.

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
