# Peer review of "Decoupling the response of an estuarine shrimp to architectural components of habitat structure"

_PeerJ, doi:10.7717/peerj.2244_

## Round 0.1 · original submission · Major Revisions

While the reviewers have differed in their recommendation for this paper, both have questioned the novelty of the work and, more importantly, have raised concerns regarding the extrapolation of the findings to natural systems. Both have also questioned some of the conclusions drawn from these observations and suggested a number of potential revisions, which, should you choose to resubmit, I would ask you to adhere to closely.

Reviewer 1 ·

Basic reporting

Ambiguous text here and there (see general comments below) and lack of accuracy in describing results using terms like "very close" or differences were "evident". These are not very scholarly ways of describing results.

Experimental design

The methods are poorly described often without any rationale being offered to justify particular choices. For example, the whole experiment is done with 5 shrimps per tank. Why is that? Is 5 shrimps/10-gallon the natural density of shrimps? Without a proper rationale how can we tell that these responses are meaningful in any way? Even though there is nothing wrong with the experimental design of the first experiment, it is impossible to assess the validity of these findings in context with what happens in nature.

Validity of the findings

I am not convinced that this paper brings anything new to the field. One of the great challenges in this field is to choose/work with an artificial habitat that matches to a certain degree the natural habitat that is used by the study organism in its natural setting. Researchers go to extreme lengths to mimics those habitats, for example, taking molds of oyster shells to reproduce the complexity of oyster beds. This is not what happened here. In this study, the authors clearly admit that they have not attempted to mimic any habitat in particular. Without such critical evaluation of the "ecological" representativity of the artificial mimic, this study has very little ecological meaning.

The statistical analysis in Table 1 is incomplete (residuals term? no. of replicates?) which makes it impossible to figure out whether it is right or wrong. The inclusion of the "army man" as an anecdotal comparison achieves noting more than that. Studies using fractal dimension have shown that similar objects can have the same exact D but affect the associate fauna in different ways. Finding more shrimp in habitats that have a "red army flag" can not be attributable to Fractal complexity alone. It is not even clear how one can calculate fractal dimension of a 3D object like that.

With so many little issues I find myself wondering what is the real point of this paper.

Additional comments

Specific comments:

L70 This paragraph is redundant for the main argument of the paper.

L77-L81 If "other factors" are also important...?

L88-90 "were not deliberately constructed to resemble any particular naturally-occurring forms.." What is the point of this experiment then? I am not sure what is the point of measuring responses to a habitat that can not be compared directly with the one these shrimps find in nature.

L106 Please provide the actual dimension of each aquarium. How is this related with the typical movement distances of these shrimps?

L117 I understand the reason why these non-mimic structures were used but they still have to make some sense "ecologically". Yes, with the amounts of garbage in the ocean there must be a place where little plastic soldiers are "natural" habitat for marine organisms. The authors offer explanation rationale for this choice and particularly for why they didn't use other mimics. This hypothesis could have been tested with other mimics like boulders or other type of macrophyte-like mimics (e.g. seagrasses).

L129 What is a "Randomized Complete Block"? The authors can not possibly expect that reader know what this is. Please avoid jargon like this and specify what are different combination of treatments.

L137 Why five? Do these densities resemble any sort of natural densities? Does shrimp density in itself influence their response to habitats structure?

L147 "choice"

L152 The fractal analysis seems to come from nowhere. I wonder if this is an afterthought and was a posteriori.

L160 Why would you do this? The normal procedure is to log(x+1). This is not a standard statistical procedure so it should be backed up with references.

L180 replace by "Shrimp were found consistently"

L183 remove "evident" as it can not be supported by the data. The averages can be greater or smaller despite being significant ror not.

L185 remove "very close". This is not appropriate description of the results.

L187 The authors should add the "amount of material" to the table in figure 1.

L189 These conclusions are completely confunded. The assertion that these non mimic effects are

L193 How is this relevant? You are comparing different materials, colours, shapes. It can be attributed to any of these differences.

L195 This result is quite extraordinary even if it has no sort point of comparison with reality. A perfect correlation is always a great thing to have but how is this related with the range of "habitat" amounts available for shrimps to chose from.

L200 "with the army men treatment essentially on the curve." I am not convinced this help support your case. The "Army men" results are confounded because they might attract greater densities because they have different colours (e.g. opaque). Fractal dimension summarizes different aspects of "habitat structure" and can indeed be used to compare structures with different densities of stalks, branching, etc. Having these in the same graph adds nothing to your argument.

L240 I think there must be a clear definition of what the authors mean by "attraction" as opposed to "behaviour choice".

L242 This makes no sense. The authors showed no evidence that the "branched" structures were similar to any particular habitat that these shrimps might use in nature.

L246 "habitat choice that might differ depending on specific behaviour being displayed"... No idea what this means.

L290 "in many fundamental ways, including active and passive accumulation of individuals, amelioration of competition and environmental conditions, and predator avoidance." I am not sure this is that clear as the authors portrait it to be. At least I have found no new evidence in this manuscript to back any of the "mechanisms". Reading these conclusion on their own it seems that these are arising from this study which is not true.

L295 "one of the more robust and predictable of ecological patterns." I can not understand why the authors would chose to end with a sentence that is far from true. There are many examples showing that complexity not always increases abundances.

Figure 2 - The word "novel" should be removed as it is inconrrect. There is nothing more "novel" about "Army man" than the branched wire. None of these are true habitats. The fact that one looks more like what might be a "natural" habitat is totally artificial and the authors provide no evidence for that.

Figure 3 - How many replicates were used to calculate each SE? Without such information it is impossible to figure out whether this is right or wrong.

Figure 5 - How many replicates?

Table 1 - This table is incomplete. Where is the Residual? How many replicates were used in the analysis? What is the response variable?

Reviewer 2 ·

Basic reporting

The article has been written to a high standard, it is a fairly straightforward study and has been well structured.
I found the caption for figure 1 quite confusing and I also thought the results shown in figure 5 could be better presented (see general comments).

Experimental design

The experimental design was well planned and robust. The methods are clear and well described

Validity of the findings

Given the experimental design and statistical approach I have no issues with the quality of the data collected in this study. The conclusions seem appropriate although they do stretch things slightly at some point in the discussion. See general comments.

Additional comments

ms: Decoupling the response of an estuarine shrimp to architectural components of habitat structure
The above titled ms details a laboratory study on the response of grass shrimp to changes in the quantity and complexity of artificial structure in laboratory experiments. Overall, the ms is well written and structured, the statistical analysis are appropriate given the authors questions, experimental design and data. Results are generally well reported. Findings are interesting though given some of the literature cited, and patterns observed in numerous other field studies, not hugely novel.
My only real concern with this ms is the potential effect suddenly viewing the shrimp by lifting the plastic shield may have on the behavior of the animals. A major point made about these experiments is they were conducted in the absence of predators, yet I would imagine suddenly removing a shield from one side of the tank may elicit a response in the prawns similar to the sudden appearance of a predator? This needs to be addressed at least at some point in the discussion.

Minor Points
Ln 97 – Some more detail on why this species was chosen for the experiments would be nice. For example, is it known that this species is attracted to structural habitat? Also, could you provide the mean ± S.E. of the animals used in the experiments?
Ln 100 – Were the holding tanks subjected to the same 12:12 h light/dark regime as the experimental arena? A little more detail is required here.
Ln 236 – More detail on the sizes of the shrimp is needed back at line 101 if you are to include statements like this one.
I thought some of the discussion presented at lines 267 was a little bit of a stretch given the data presented in this relatively small and simple study. I understand the authors are linking their findings to broader applied ecological concepts and examples here, but I feel they are pushing things a little with the data presented here….
Caption for Figure 1 seems incorrect ‘four unbranched stalks had the same surface area as one unbranched stalk’? I also found the table in Figure 1 confusing, maybe I missed something but was there suppose to be the same number of unbranched and branched stalks in each test area? Where are the dashed lines referenced in the caption?
Figure 5 – I found this figure a bit of a confusing way to show what appears to be a neat result and don’t think it adds much to the ms. Could this information be better conveyed in a simple table?

---

## Round 0.2 · Minor Revisions

Dear Jeff, I would like you to consider acknowledging the limitations of active choice expts, perhaps simply saying as much and referring the reader to Underwood and Clarke, and considering the work by Loke.

Thank you for your submission, Wayne

Reviewer 1 ·

Basic reporting

The first note to make is that it is quite annoying to review a manuscript/rebuttal letter without any line numbers which just makes it more complicated to track down which (and where) changes were made to the manuscript to address particular comments.

The authors have proficiently answered most queries from the previous review. The manuscript seems much clearer following the changes.

Experimental design

I am happy to see the changes made improved the explanation of the methods but still think that the authors could have done a better job at acknowledging the limitations of their “active choice” experiments. The procedure used has several limitations that must be, at least, mentioned. I suggest reading Underwood and Clarke (2005) for reference.

A.J. Underwood, K.R. Clarke (2005) Solving some statistical problems in analyses of experiments on choices of food and on associations with habitat

The more detailed explanation of the procedure to determine the fractal dimensions is clearer but still needs to acknowledge the limitations of procedure to account for the 3-dimensionality of the objects. There are studies out there that have done this and I again suggest that the authors acknowledge the recent work by Loke and colleagues that measured (and simulated) 3D fractal dimensions.

Validity of the findings

I disagree with the authors’ assessment of the “validity” of their results but can not offer a better solution. The authors say in their rebuttal letter, that “The responses to this were very clear – it is the amount of structure and not how it is arranged that was important for shrimp response. We don’t know that this was entirely predictable, as two branched and eight un-branched stalks look qualitatively quite different (to our eyes at least). We believe our results should contribute to the ecological literature.” Just because, apparently, no one else has suggested this before can not be an argument for a paper to make a contribution to the field. There are other recent (and not cited) studies suggesting this or very similar things. For example, Matias et al 2015 PlosOne showed that spatial arrangement (keeping total amount fixed) does not significantly affect the assemblages of macroinvertebrates. Also, it is having a look at a recent paper by Loke et al 2015 Ecology which arrives to the opposite conclusion: if you keep habitat amount fixed, structural complexity (measured using Fractal Dimensions) has an important effect on assemblages. I think you should consider including these studies (and possibly others by Loke) in your study.

Finally, I do not agree with the justification regarding the lack of “ecological context” but accept that the authors are right in saying that there are many others papers on this subject that are completely “detached” from a particular natural habitat - that does not make it right to do it though ;) The authors are free to pursue any avenue of research but they must be, in my opinion, asked/compelled to make links to natural habitats somehow. The authors seem to think that that is not the way to go, which in my opinion reduces the paper's potential reach and appeal.

---

## Round 0.3 · accepted · Accept

Jeff - thanks for your submission to PeerJ. Wayne